# When the Going Gets Tough and the Environment Is Rough: The Role of Departmental Level Hostile Work Climate in the Relationships between Job Stressors and Workplace Bullying

**DOI:** 10.3390/ijerph20054464

**Published:** 2023-03-02

**Authors:** Lena Zahlquist, Jørn Hetland, Guy Notelaers, Michael Rosander, Ståle Valvatne Einarsen

**Affiliations:** 1BI Norwegian Business School, Kong, Chr. Frederiks gate 5, 5006 Bergen, Norway; 2Department of Psychosocial Science, University of Bergen, Christiesgate 12, 5020 Bergen, Norway; 3Department of Behavioural Sciences and Learning, Linköping University, 581 83 Linköping, Sweden

**Keywords:** role conflict, workload, hostile work climate, workplace bullying

## Abstract

In line with the work environment hypothesis, the present study investigates whether department-level perceptions of hostile work climate moderate the relationship between psychosocial predictors of workplace bullying (i.e., role conflicts and workload) and exposure to bullying behaviours in the workplace. The data were collected among all employees in a Belgian university and constitutes of 1354 employees across 134 departments. As hypothesized, analyses showed positive main effects of role conflict and workload on exposure to bullying behaviours. In addition, the hypothesized strengthening effect of department-level hostile work climate on the relationship between individual-level job demands and individual exposure to bullying behaviours was significant for role conflict. Specifically, the positive relationship between role conflict and exposure to bullying behaviours was stronger among employees working in departments characterized by a pronounced hostile work climate. In contrast to our predictions, a positive relationship existed between workload and exposure to bullying behaviours, yet only among individuals in departments with low hostile work climate. These findings contribute to the bullying research field by showing that hostile work climate may strengthen the impact of role stress on bullying behaviours, most likely by posing as an additional distal stressor, which may fuel a bullying process. These findings have important theoretical as well as applied implications.

## 1. Introduction

Over the last decades, a growing body of workplace bullying research has thoroughly documented its detrimental consequences for those exposed, yet also for bystanders and even for organizations and societies at large [1,2,3]. Targets of workplace bullying tend to suffer a range of mental and physical health problems, negative job attitudes, and increased intentions to leave their employment [4,5]. In addition, organizations and societies may suffer direct economic losses, due to reduced productivity, lowered workability, and increased health problems among all those involved [2,6]. In order to prevent workplace bullying and its damaging consequences, there is a need to understand potential risk factors, as well as maintaining and escalating factors and the possible social mechanisms involved, as workplace bullying is a complex process theorized to be influenced by a range of person-related, work-related and contextual factors [7].

To date, the work environment hypothesis [8,9] has been the prevailing overarching theoretical framework for studying antecedents of bullying. The hypothesis claims that a poorly organized and stressful work environment may lead to bullying by creating stress, frustration, and conflicts among employees, often in combination with a lack of adequate management interventions [10], and in situations where there is a prevailing hostile climate in the working group [11]. In support of this proposition, work-related stressors and the social climate are found to be the most robust predictors of workplace bullying, in particular job demands in the form of role stressors and heavy workloads [12,13].

Heightened levels of stress from taxing job demands may lead some employees to misbehave and break the social norms of polite behaviour, fuelling interpersonal conflicts, as also proposed by the social interactionist perspective on aggression [14,15]. Furthermore, being under such stressors may increase one’s vulnerability in negative interpersonal relationships, which again may lead to a more negative perception of one´s social working environment.

Although the link between such strains and exposure to bullying has been firmly substantiated [16,17], the main proposition and novel assumption investigated here is that there may also be specific contextual and departmental conditions, e.g., a hostile climate, under which such frustration caused by perceived stressors will be even more likely to result in spiralling interpersonal conflicts and increased vulnerability in targets-to-be. Based on the conservation of resources theory, we may see such a hostile climate as a resource passageway: organizational environmental conditions that detract, undermine, obstruct, or impoverish people’s or group’s resource reservoirs [18,19], which will increase their stress levels, decrease their resources and hence increase their vulnerability.

The present study therefore adds to the literature by testing the propositions in the work-environment hypothesis, namely that the risk of being exposed to bullying is higher when under the influence of psychosocial stressors and particularly so when working in a general hostile working climate where interpersonal conflicts and aggression flourishes. Such a combination of stressors would indicate a perceived demanding work situation which creates stress in the focal person who is also faced with being in a demanding social context characterized by a lack of normal social resources. Thus, working in a social context plagued with interpersonal conflicts, aggression, and hostility may be a taxing demand, yet it also denotes a lack of the ordinary social resources of friendship and social support generally present and available to employees. Following the conservation of resource theory, this would imply that there may be a multiplying effect of situational and contextual demands [19,20]. In addition, this happens in a situation with a loss of contextual resources that would potentially help ones coping with these demands.

Our main assumption to be tested is therefore that job demands such as role stress and taxing workloads are risk factors for exposure to bullying, and particularly so when being in a hostile working climate. In this, the present study has important theoretical, methodological, and applied contributions. Theoretically, we contribute by being the first to test an important proposition in the work environment hypothesis, while also showing how factors at different levels of analysis may interact to heighten the risk of exposure to workplace bullying. In terms of methodology, we contribute by employing a multilevel design in line with the theoretical assumptions. From an applied perspective, we contribute with nuanced information on how to prevent and manage bullying at work.

### 1.1. The Concept of Workplace Bullying

Workplace bullying is about the systematic and ongoing exposure to mistreatment and harassment by one’s colleagues or superiors, which may become “an escalating process in the course of which the person confronted ends up in an inferior position and becomes the target of systematic negative social acts” [21]. These negative social acts constituting workplace bullying can take different forms, and either be work-related, such as the withholding of information that affects the target’s work performance, having key areas of responsibility removed or replaced with more trivial or unpleasant tasks, or they can be person-related, such as gossip and rumours about you being spread or being the target of spontaneous anger [22], often also including acts of social exclusion or non-inclusion. Traditionally, research on workplace bullying has a focus on the target, who may be exposed to such acts from a range of sources and perpetrators, where the total exposure is at the heart of the experience. Furthermore, being a gradually escalating process, exposure to workplace bullying has been shown to manifest itself in low as well as high intensities at any given time-point [22,23]. Since we are interested in understanding the risk factors associated with such bullying, the present study will investigate the whole range of experienced exposure to bullying, from low intensity unwanted negative acts to full-blown cases of bullying; conceptualized as exposure to bullying behaviours.

### 1.2. Antecedents and Risk Factors of Exposure to Workplace Bullying

When examining the antecedents of workplace bullying, most of the research has focused on more proximal work-related factors experienced directly by individuals, such as the extent employees experience role conflict or high and taxing workloads, as well as the perceived leadership style of one´s immediate superior [10]. However, contextual risk factors may exist on different levels of the organization, e.g., in the form of a hostile work climate in the department as a macro level stressor. However, the risk factors proposed in the work environment hypothesis [11] tends to be tested as independent risk factors. However, a central and novel assumption in the present study is that risk factors at different levels may interact to reinforce the risk of individual exposure to bullying. So far, such mechanisms are relatively poorly understood [24]. Hence, we propose that a hostile work climate in the department constitutes a resource passageway, or rather, the absence of an important resource functions as a stressor that will influence other job demands [25], increasing the risk of exposure to bullying behaviours at the individual level. Following the work environment hypothesis and the conservation of resources theory, we propose that working in a department with a hostile work climate may boost the job demands—bullying relationship in several ways. Firstly, working in a hostile climate may in and of itself serve as an environmental stressor for employees which may come on top of other stressors [26,27]. In a hostile work climate, employees may also experience less social support and less constructive intervention by bystanders such as their peers and superiors [28], leading to a lack of important buffering mechanisms. Being in such an environment may thus also reduce and negatively affect one’s coping resources [29]. Secondly, a hostile work climate may trigger similar reactions and negative treatment from a range of peers who all would react to the ambient stressors crated by the hostile working climate and particularly against stressed-out targets in that environment. This will not only amplify the exposure to negative treatment from many sources, but also put the target in a more inferior position of being less able to defend themselves and more prone to perceive social interaction as negative and unwanted. Additionally, perpetrators-to-be may “watch and learn” patterns of interpersonal misbehaviour from their colleagues when they are frustrated by others [30], providing a breeding ground for destructive behaviour and interactions in the workplace and particularly so when under the influence of stressors [29].

#### 1.2.1. Role Conflict

Role stressors, and especially role conflict, is one of the most studied work-related risk factors of workplace bullying and has consistently been found to be among the strongest predictors of perceived bullying [8,12,31]. Role conflict can be described as the simultaneous existence of two or more sets of expectations toward the same person, such that compliance with one makes compliance with the other difficult [32,33]. Experiencing high degrees of conflicting expectations and demands from leaders or colleagues is found to be associated with elevated levels of stress and frustration, as it may hinder efficient goal attainment at work [34]. Holding different and incompatible roles may also create frustration in both the focal person and other role-senders in the working environment, with a risk of conflict escalation and negative reactions toward the focal person.

In line with the work environment hypothesis, Einarsen and colleagues [8] argued that the association between role conflict and workplace bullying is due to the creation of strain and frustration in the working group, which may then escalate into harsh conflicts and potentially bullying. Along similar lines of reasoning, it has been argued that role stressors may act as ambient stressors that are perceived not only by victims, but also by perpetrators. While perpetrators may enact bullying in response to those stressors [12], they may also retaliate against stressed-out colleagues who may violate the social norms of polite interaction and as a response to the role conflicts of the focal person. Hence, role conflict may fuel escalating conflicts among colleagues, in line with a social interactionist approach to aggression [35]. Such explanations were supported in a longitudinal study by Balducci and colleagues [36], in that role conflict positively predicted both being bullied and bullying enactment. A representative study of Norwegian workers [37] also documented how high levels of role conflict are reported by self-reported bullies. This further aligns with the frustration-aggression hypothesis and the social interactionist perspective, which states that aggression is elicited by negative stressful events [38], by affecting both future perpetrators and targets. Hence, the following hypothesis is proposed:

**Hypothesis** **1.**
*There is a positive relationship between role conflict and reported exposure to bullying behaviours at work.*


#### 1.2.2. High Workload

In addition to role conflict, experiencing high workload has been suggested as an important precursor of bullying [31,39]. Although not as consistent as the research findings on role conflict, studies from a variety of countries do point to a relationship between workload and exposure to bullying behaviours [31,39,40]. In the present study, the term workload can be described as the amount and speed of work to be performed, which determines whether you need to work fast or extra hard to get your tasks done [41]. While role conflict is a clear example of a hindrance demand, workload may however be seen as a challenge demand according to the challenge stressors-hindrance stressor framework [42], a fact that may account for the less robust findings in the literature regarding risk of exposure to workplace bullying. However, in line with the work environment hypothesis, a high workload over time, and especially without sufficient resources, may result in strain and conflict escalation, finally resulting in bullying [39,43]. This may either be the result of the workload serving as an ambient stressor, affecting both targets and bullies, or that employees who experience particularly high workloads become stressed out, therefore becoming more vulnerable and acting in ways that irritate and annoy colleagues and superiors, hence further triggering or fueling the bullying process [8,24,44]. Besides, being exposed to high workloads over time is argued to be a risk factor for conflict escalation, since those involved have sparse time and limited resources for conflict resolution [45]. Thus, we do propose:

**Hypothesis** **2.**
*There is a positive relationship between workload and reported exposure to bullying behaviours at work.*


#### 1.2.3. Hostile Work Climate

In one of the pioneering studies on antecedents of workplace bullying and in support for the work environment hypothesis, Einarsen and colleagues [8] found that a poor social climate at work was one of the factors that proved to be most strongly associated with bullying, along with role stressors. In the present study, we employed the concept of a hostile climate, which refers to a social environment in the department characterized by escalated interpersonal conflicts and aggressive behaviour. However, such a climate may be more than a mere risk factor. Frustrated, insecure, and stressed out employees will often look for support in his or her immediate work environment [46], as these are normally important resources that may alleviate the effect of a given stressor. Hence, a potentially important factor in predicting whether bullying will occur at the individual-level is the group context in which people may actively condemn bullying behaviours, do nothing to stop it, allow it or even encourage or normalize such behaviours. People tend to seek information from the social context surrounding them when it comes to behaviours and making choices [47]. Employee perceptions of the working group’s norms, practices and procedures regarding social interaction can therefore have a significant impact on how employees react to stress, as this may function as a frame of reference for acceptable behaviour in stressful situations [30]. Destructive employee behaviour can thus be more likely to occur if such behaviour is “common practice” in the work environment [30], e.g., when interacting with colleagues stressed out by conflicting demands and expectations. Hence, in departments with a hostile work climate, where the interaction between colleagues is permeated by conflicts and aggression, there might be increased risk for ongoing interpersonal frustration to evolve into aggression and bullying behaviours. In addition, stressed out employees may become more vulnerable and have less resources to defend oneself in a hostile working climate. An ambient hostile climate consisting of a range of escalated conflicts and aggressive outlets in the department may in and of itself be stressful and create uncertainty. It may be seen as a resource passageway, which is an organizational environmental condition that detracts, undermines, obstructs, or impoverishes the people’s or group’s resource reservoirs [18,19]. Thus, such a hostile environment will also lack resources in the form of social support from peers and superiors.

To our knowledge, only two studies have tested organizational climate as a moderator in the antecedent–bullying relationship, employing the concept of conflict management climate [48] and high-performance work practices [20]. In the first study, the construct of the conflict management climate was investigated at the group-level and found to have a buffering effect on the relationship between job demands and exposure to bullying behaviours [48]. A conflict management climate was defined as employees having confidence that conflicts will be properly managed and resolved, as the organisation and its managers have proper procedures and routines for constructive conflict management [49,50]. In line with the conservation of resources perspective, a strong conflict management climate served as a resource passageway which buffered demands at an individual level, as it presumably led to an increased sense of control and available social resources, probably in combination with effective management interventions [48,50]. In the second study, the construct of high-performance work systems was modelled at the organizational level with the idea that it would act as a resource passageway. These systems buffered the effect of role conflict on workplace bullying, as it presumably led to a better use of both job and personal resources [20], as employees could draw on these contextual resources to replenish resources that were depleted [51].

Instead of providing such organisational resources, a hostile work climate acts as a demand [25], in that interpersonal conflicts and aggressive behaviour flourishes in the department, hindering the social support people need when exposed to stress, and serving as an additional stressor when exposed to stress because of job demands, thereby strengthening, boosting or increasing the effect of the latter [29]. Therefore, in line with the work environment hypothesis and relying upon the notion of resources passageway in the conservation of resources theory, we believe that a department-level hostile work climate will increase the stress and interpersonal frustration and conflict arising from high job demands [i.e., role conflict and workload), while also reducing the availability of social resources when faced with these demands, subsequently fuelling the bullying process.

Since the concept of organizational climate has been described as the aggregated perceptions of group members regarding a particular aspect of the work setting [52,53], we will apply a multilevel design with group-level perceptions of hostile work climates. Due to the lack of multilevel research needed to address antecedents at the group-level [54,55], the role of the organizational climate in strengthening workplace bullying remains underdeveloped in current research and theory [56]. Given that most organizations are hierarchically structured with systems of social interactions affecting individuals, a multilevel design is essential [57]. In addition, gaining more knowledge regarding the group-level of analysis may also have important practical implications when it comes to developing appropriate organizational policies and interventions in groups and departments [48].

Hence, the present study first replicated findings on the relationship between two individual-level predictors of bullying (i.e., role conflict and workload) and reported exposure to bullying behaviours. Yet, the main aim is to test the hypothesis that these relationships depend on a department-level hostile work climate as a contextual factor. By integrating department-level hostile work climates as a moderator, we aspire to test, extend, and potentially provide additional validation of the work environment hypothesis, by obtaining a more nuanced and better understanding of the antecedents and mechanisms involved in the workplace bullying process. In this, we also contribute to the general request for research on moderators in the job demands – bullying relationship [58], as well as the request to empirically investigate the effects of the organizational climate in relation to workplace bullying [24]. Working in a hostile climate will make stressed employees more vulnerable to bullying, not only by eliciting more negative acts from any given colleagues or superiors but also by creating exposure from more sources and consequently even putting the target in a more inferior position, hence creating a situation even more in line with the definition of workplace bullying. A hostile working environment may further restrain the social support an employee would normally receive when in a stressful situation.

In order to test our hypotheses, we have chosen an academic context, as it represents a competitive and complex environment which in itself may be a distal risk factor for workplace bullying [59], often also described as borderless work, which is subsequently associated with high job demands [60,61]. Hence, we propose the following:

**Hypothesis** **3a.**
*The positive relationship between role conflict and exposure to bullying behaviours is moderated by a hostile work climate. Specifically, the relationship between role conflict and exposure to bullying behaviours is stronger among employees working in departments characterized by a pronounced hostile work climate.*


**Hypothesis** **3b.**
*The positive relationship between workload and exposure to bullying behaviours is moderated by a hostile work climate. Specifically, the relationship between workload and exposure to bullying behaviours is stronger among employees working in departments characterized by a pronounced hostile work climate.*


## 2. Method

### 2.1. Procedure and Participants

The data were collected among all employees at a Belgian university in 2013 by a statistical consulting agency that specializes in the measurement of occupational stress for a Belgian Health and Safety Executive, providing us with anonymous data for the present study. These external prevention services are by Belgian law entitled to guide organizations and employers with respect to their prevention policies regarding safety, ergonomics, health, and well-being. The response rate was 48.8% and the total sample consisted of 1354 employees working in 134 departments and equivalent work units. All these units are formal scientific departments and formal administrative and technical units. Within these units, there may of course be more informal smaller teams and work groups. Yet, this reflects the official organisational departments and units of the University. 

We only retained departments consisting of over 3 respondents to secure a reasonable measure of a department level hostile climate and to reduce the risk of having only targets or perpetrators rating the climate. This resulted in the omission of 26 departments. Hence, the final sample consisted of 1290 employees within 108 departments. The size of the retained departments varied from 4 to 54 people with an average of 12. The sample is heterogenous in terms of tasks, professions, roles and organisational structures in different parts of the university, yet therefore also representative for a typical University. Forty-six percent were administrative or technical personnel, 11% were extra-ordinary academic personnel, 16% were PhD or postdoc students funded by a research fund, 19% were professors (assistant, associate or full) and finally, 8% were research- and teaching assistants. Forty-seven percent of the participants were male (53% female), with the following age distribution: 5% were under 25 years, 36% had ages between 25 and 34 years, 24% between 35 and 44, 22% between 45 and 54, and 13% were over the age of 55. Approximately 28% of participants held a managerial position and 79% worked full-time. 13% had a tenure of maximum 1 year, 33% have worked between 1 and 4 years at their current employer, 15% between 5 and 9 years, 24% between 10 and 24 years and 15% have worked for the same employer for over 25 years.

### 2.2. Instruments

To measure exposure to bullying behaviours we used the Short Negative Acts Questionnaire [62], which consists of nine items from the full version of the Negative Acts Questionnaire-Revised [63]. The items followed an introductory text stating: “How many times have you been the target of the following behaviours during the last six months?” Example items are: “Someone withholding necessary information so that your work gets complicated”, “Gossip and rumours about you”, and “Social exclusion from co-workers or work group activities”, with response categories ranging from 1 (*never*) to 4 (*once a week or more*). The scale showed good reliability, Cronbach’s α = 0.86.

The measurement of role conflict is based on four items from the Short Inventory to Monitor Psychosocial Hazards [64]. The items are: “Do you receive contradictory instructions?”, “Do you have to do your work in a way which differs from the method of your choice?”, “Do you have conflict with your colleagues about the content of your tasks?” and “Do you have conflict with your boss about the content of your tasks?”, with response categories ranging from 1 (*never*) to 4 (*always*). The scale showed acceptable reliability, Cronbach’s α = 0.78.

The measurement of workload is based on three items from the Short Inventory to Monitor Psychosocial Hazards [64]. The items are: “Do you have to work extra hard in order to complete something”, “Do you work under time pressure?” and “Do you have to hurry?”, with response categories ranging from 1 (*never*) to 4 (*always*). The scale showed very good reliability, Cronbach’s α = 0.89.

Hostile work climate was measured using four items from the Short Inventory to Monitor Psychosocial Hazards [64]. The overall starting sentence was: “How often have you been confronted with the following… during the last six months?”. The items are: “…aggressiveness from colleagues?”, “…aggressiveness from your boss?”, “…conflicts with your colleagues?” and “…conflicts with your boss?”, with response categories on a scale from 1 (*never*) to 4 (*always*). Hence, a hostile work climate on the department level is a measure of the extent that all employees in the department report to be involved in interpersonal conflicts and being faced with aggression from co-workers and superiors. The scale showed acceptable reliability, Cronbach’s α = 0.71. Prior to the multilevel analysis, the items were computed into a sum score, and a departmental average score was used at the between-level in the analysis.

### 2.3. Analyses

To utilize the multilevel structure of the data, implying that individual scores (level 1) were nested within departments (level 2), we conducted multilevel analysis using MLwiN 3.01. In the analysis, level 1 predictors were centred on the team mean, while level 2 predictors were centred on the grand mean. In order to test our hypotheses, we ran five models predicting exposure to bullying behaviours. In the first step, we ran a null model where the intercept was included as the only predictor. In step two, we tested a main effect model by adding the hypothesized level 1 predictors (i.e., role conflict and workload). In step three, in order to examine possible random effects of the level 1 predictors on the higher level (level 2), we allowed the slopes of the relationships between the predictors (i.e., role conflict and workload) and the outcome (i.e., bullying behaviours) to vary randomly. In step four, we added the hypothesized level 2 predictor (hostile work climate), explaining level 2 variance in individual exposure to bullying behaviours. Finally, in step five, we tested the hypothesized cross-level interactions between hostile work climate and the two level 1 predictors by including their respective interactional effects. Additional simple slope tests for hierarchal linear models were conducted to examine if the slopes in the potential cross-level interactions are significantly different from zero [65]. In the simple slope test, the predictors and moderators are tested at ±1 SD, and calculations are based on the asymptotic covariance matrix from the respective multilevel models using R version 3.4.3.

### 2.4. Research Ethics

The data were collected by an electronic survey distributed to employees’ e-mail. Participation was voluntary. No members of the surveyed organization or the Health and Safety Executive had access to any questionnaires, herewith guaranteeing anonymity. E-mail addresses were deleted. Thereby the statistical agency met with the Belgian data protection regulations. Respondents were informed about the purpose of the research and that choosing to participate would indicate their informed consent.

## 3. Results

### 3.1. Descriptive Statistics

Table 1 shows the means, standard deviations, inter class correlations (ICC1/ICC2), and within- and between-level correlations for all study variables. Correlational analysis showed that at the within-level, significant positive correlations existed between the two job demands and exposure to bullying behaviours, respectively, with the strongest relationship between role conflict and exposure to bullying. Furthermore, role conflict was positively related to workload. On the between-level, strong positive correlations exist between hostile work climate and bullying, workload, and role conflict.

### 3.2. Multilevel Analysis

Table 2 presents the results from the multilevel analysis predicting exposure to bullying behaviours. The null model revealed significant variance components on both levels ($\varepsilon_{0ij}\;$= 0.193, *p* < 0.001;$\;\mu_{0j}$ = 0.040, *p* < 0.001), where 83% of the variance in bullying behaviours exists at level 1 and 17% at level 2 of the analysis. In Hypotheses 1 and 2, we postulate that the two level 1 predictors (role conflict and workload) positively relate to bullying behaviours. The results from the main effect model revealed a significant positive relationship between role conflict and bullying behaviours (B = 0.474, *p* < 0.001) and between workload and bullying behaviours (B = 0.071, *p* < 0.001). Thus, both Hypotheses 1 and 2 were supported.

In order to obtain the correct standard errors for potential cross-level interactional effects, the higher level random slopes for both predictors were estimated in the next model [66]. As can be seen in Table 2, the random slope of the role conflict–bullying relationship was significant ($\mu_{1j\;}$ = 0.035, *p* < 0.01), while the corresponding random slope for the workload–bullying relationship was not significant ($\mu_{2j}$ = 0.000, n.s.). This suggests that only the relationship between role conflict and bullying systematically differs across departments, while this is not the case for the relationship between workload and bullying. Moreover, introducing our level 2 predictor hostile work climate revealed a strong and significant association with individual level exposure to bullying behaviours at the higher level (B = 0.955, *p* < 0.001).

In Hypotheses 3a and 3b, we expect that a hostile work climate on level 2 positively moderates the positive links between the two level 1 predictors (role conflict and workload) and bullying behaviours. The positive interactional effect between role conflict and department-level hostile work climate in the prediction of bullying behaviours was significant (B = 0.981, *p* < 0.001). However, contrary to our expectations, the interaction model revealed a negative interactional effect between workload and department-level hostile work climate in the prediction of bullying behaviours (B = −0.420, *p* < 0.001). In order to visually inspect if the pattern of the interactional effects were in accordance with our hypothesis, we plotted the slopes of the interactional effects in Figure 1 and Figure 2.

Figure 1 provides a visualization of the significant interaction effect between role conflict and department-level hostile work climates. As seen in the figure, and in accordance with Hypothesis 3a, there is a stronger positive association between role conflict and bullying behaviours among individuals working in departments characterized by a hostile work climate, as compared to individuals working in departments characterized by low levels of hostility. Despite these differences, a formal test of the slopes at ±1 SD of the moderator revealed a significant slope both for those in a high hostile work climate department (Slope = 0.620, *z* = 17.22, *p* < 0.001) and for those working in a low hostile work climate department (Slope = 0.288, *z* = 7.50, *p* < 0.001). Inspection of Figure 2 reveals, surprisingly, that in departments characterized by high hostile work climate, the level of reported bullying behaviours is independent of the experienced workload. Correspondingly, the simple slope test revealed that the relationship between workload and bullying behaviours was not significant in the departments with a high level of hostile work climate (slope = −0.011, *z* = 0.41, n.s.). Still, the figure shows that more exposure to bullying behaviours are reported in departments characterized by a hostile work climate, independent of the experienced workload. In contrast, a clear positive relationship exists between workload and exposure to bullying behaviours among individuals in departments with low hostile work climates. Accordingly, the simple slope test reveals a significant positive slope among those in departments with a low hostile work climate (slope = 0.131, *z* = 5.29, *p* < 0.001). In summary, Hypothesis 3a was supported, while the multilevel analysis did not yield support for Hypothesis 3b.

In order to rule out the possibility that the relationships can be explained by relevant third variables, we ran all the analyses while controlling for gender, age, and tenure. However, the analyses showed that none of the control variables significantly predicted exposure to bullying behaviour. Based on this, we decided to only report the most parsimonious analyses excluding the control variables, in line with the suggestions of Cohen [67].

## 4. Discussion

The aim of the present study was to extend our understanding regarding work-related antecedents of workplace bullying, by investigating the interaction of potential risk factors at different organizational levels. Based on the work environment hypothesis and the social interactionist approach to aggression, we hypothesized that experiencing high levels of role conflict and workload would be positively related to exposure to bullying behaviours at work. Job demands, such as role conflict and workload, have consistently been found to predict self-reported exposure to bullying behaviours in the workplace. Based on research on the notion of resource passageways in the conservation of resources theory, we further examined whether the relationship between these job demands and exposure to bullying behaviours would be strengthened by working in a department characterized by a pronounced hostile work climate, that is, a climate where escalated interpersonal conflicts and aggressive outlets prevail in the social environment.

As hypothesized, the results of the analyses showed positive main effects of individual level role conflict and workload on exposure to bullying behaviours, with the strongest relationship with bullying exposure existing for the former. Hence, employees who experience elevated levels of role conflict and workload tend to report more exposure to bullying behaviours. Accordingly, the findings support the work environment hypothesis [8,9], which claims that bullying is related to stressors in the psychosocial work environment that create stress, frustration and conflicts among employees. Yet, they are also in line with a social interactionist perspective on aggression in that such aggressive outlets may follow from retaliation and aggressive outlets from perpetrators against stressed out and vulnerable targets [14,15]. Being exposed to high job demands over time, without sufficient resources, is related to negative outcomes such as sleep problems, fatigue, and impaired health [68]. These indirect health effects, as well as the direct stress triggered by role conflict and high workload can, according to the social interactionist perspective, lead to behavioural changes, such as the violation of social norms, which may provoke frustration and aggressive behaviour from colleagues, subordinates and superiors, who then may target the stressed-out employee [9,15,69]. The results of the present study aligns with several previous studies, showing that employees who experience high levels of role conflict and workload are more often exposed to bullying behaviours [13].

Furthermore, the results showed that the positive relationship between role conflict and bullying behaviours was stronger for employees working in departments with a pronounced hostile work climate. Hence, the present study is, to our knowledge, not the first that supports the notion of resources passageways, but it is the first to empirically demonstrate the strengthening effect of a hostile work climate on the link between role conflicts and exposure to bullying behaviours. Although several previous studies have shown an association between poor organizational climate and exposure to bullying [55,70,71], knowledge regarding the potential intervening effect of the organizational climate, in combination with other stressors, is scarce [72,73]. Despite the fact that organizational climate has been little investigated in the bullying literature, a long-held proposition in the work environment hypothesis is that the risk of exposure to bullying will be high in departments with hostile work climates [8,9,21]. Considering this, the present study makes an important theoretical contribution by providing this additional validation of the work environment hypothesis, showing the interactional effects among its proposed risk factors. Yet, as a hostile climate did not strengthen the relationship between workload and exposure to bullying, this also provides some important nuances in this overarching proposition. In fact, and in line with previous studies, the results indicate that a high workload is a risk factor for exposure to bullying in normal social climates. Yet, high workload is not a risk factor when in an ambient hostile climate.

The strengthening effect of a hostile work climate in relation to role conflict may be explained in several ways. First of all, and in line with the work environment hypothesis, a hostile work climate may serve as an additional distal stressor, interacting negatively with other work-related stressors [74]. In departments where the interaction between colleagues is permeated by conflicts and aggression, employees are likely to have poorer social relations. Studies have shown that employees who lack social support from their colleagues tend to cope less effectively in response to stressful situations [75,76], making those who work in hostile climates more likely to experience their work-related stressors as demanding, taxing one’s resources. In a study by Mawritz and colleagues [29], employees working in hostile climates had a tendency to cope with their environment by psychologically withdrawing, which is hampering the replenishment of resources. Such withdrawal may then cause employees to not intervene or make their voice heard when mistreatment and unfairness is taking place at work. Hence, it impoverishes the resource reservoirs of employees and groups. Subsequently, if bullying incidents go unchecked, there is a heightened risk of bullying behaviours becoming “normalized” [77]. In a department that lacks inhibiting norms against such behaviour, the threshold for frustration to turn into aggression and bullying behaviours may also be lowered, an assumption in line with the social information processing theory [47]. Along similar lines of reasoning, the concept of emotional contagion [78], described as the tendency to mimic the verbal and behavioural aspects of another person’s emotional expression [79], may provide an additional explanation for why bullying behaviours can be a result of, and spread in, a hostile work climate [80], particularly when under the influence of other stressors. These kinds of tendencies have been documented, for instance in a study by Robinson and O’Leary-Kelly [30], who found that individuals’ antisocial behaviours at work were shaped by the antisocial behaviour of their co-workers. More recent studies have also shown that if leaders act aggressively, this may have a strong impact on their employees’ behaviours [74].

Finally, and contrary to our expectations, the present study results showed no significant strengthening effect of department-level hostile work climates on the relationship between workload and bullying behaviour. Hence, our hypothesis that the relationship between workload and exposure to bullying behaviours would be stronger among employees working in departments characterized by a pronounced hostile work climate was not supported. Still, the results clearly show that more exposure to bullying behaviours are reported in departments characterized by a hostile work climate, independent of the experienced workload. Further, and in contrast to our predictions, a positive relationship between workload and exposure to bullying behaviours was found among individuals in departments with a low hostile work climate. However, the analyses revealed that the random slope for the workload–bullying relationship was not significant, which suggests that the relationship between workload and bullying did not systematically differ across departments. This means that any interpretation of these results should be done with caution. Still, if we are to try and interpret these findings, it may be that in departments characterized by a pronounced hostile work climate, the environment is already so negative and stressful that whether the workload is high does not really matter. On the other hand, in departments with low levels of hostility, there is an increase in exposure to bullying behaviour among those who experience high workload, a finding that is in line with our second hypothesis.

If we further compare the two studied job demands, role conflict and workload, they are considered to be different kinds of stressors in the literature [42,81]. While role conflict is considered to be a hindrance demand or a “bad” stressor that inhibits an employee’s ability to achieve valued goals, workload is termed by some as a challenge demand or a “good” stressor, with the potential to promote personal growth and achievement [82]. This distinction between the very nature of the studied stressors may be one explanation for why role conflict and workload seem to have somewhat different effects in a hostile work climate. Yet, these issues still need to be further investigated.

### 4.1. Practical Implications

We believe that the present study has several important practical implications for leaders and HR personnel working to prevent workplace bullying. Firstly, the results of the present study show that employees who report high levels of role conflict and workload are more prone to be exposed to bullying behaviour, regardless of whether they work in a department with a hostile climate or not. This stresses the importance of having well-organized working conditions, in order to reduce conflicting roles and to strive to make sure that employees have sufficient resources in periods of high workload.

Secondly, managers should pay close attention to the organizational climate. Although it is known that the organizational climate is a driving force in organizational behaviour [83], the present study sheds light on how a hostile work climate may serve as a catalyst in a stressful environment, and together with other risk factors may increase the risk of bullying behaviours taking place. When it comes to shaping the organizational climate in a working group or department, this will to a great extent depend on the leadership style and supervisory practices, as leaders have the responsibility for the work environment and the power to influence and develop the organizational climate, through both their expectations and standards of behaviour [84,85]. The organizational climate is based on employees’ perceptions of the policies, practices, and procedures as in a climate of conflict management, and the behaviours they observe being accepted, rewarded, and encouraged [83,86], as in the case of the hostile working climate in the present study. Hence, it is by shaping these aspects that the organizational climate can be changed.

A recent study by Dollard and Bailey [85] showed that the organizational climate, which in their case was a psychosocial safety climate (PSC) that has been shown to be a salient organisational level predictor of bullying, can be shaped through formal and structured interventions. Training middle management to enact PSC in work-units increased PSC within a 4-month period. A similar climate construct, the conflict management climate, has also been found to have a preventive effect on bullying at a team-level [48]. In a recent prospective study, Hamre and colleagues [87] showed that, by creating a strong conflict management climate in which employees perceive and trust that interpersonal problems are firmly and fairly managed, the escalation of new and existing bullying cases may be prevented. Finally, taking a multilevel approach by investigating the organizational climate at a group-level may also have practical implications, as intervention programs directed at the group-level are found to be more effective than those directed solely at the individual level [88,89].

### 4.2. Strengths and Limitations

As the present study employed a multilevel design, it aligns with the theoretical foundation of the concept of climate, defining organizational climate as organizational members’ shared perceptions of the workplace [90]. Yet, the cross-sectional design with only one measuring point limits our conclusions regarding the direction of causality among the studied individual level variables. Being bullied may have worsened the employees’ roles and work tasks, as well as their perceptions of the organizational climate. However, in a study by Skogstad and colleagues [69], both bullied and non-bullied employees reported a poor interpersonal work environment in their department. Additionally, employing a true prospective design, Reknes and colleagues [91] found role conflict to predict subsequent exposure to workplace bullying.

Two criteria should ideally be fulfilled for an organizational climate structure to be appropriately captured [83]. Statistical procedures should then be conducted to aggregate the data to the organizational level of analysis [92], as done in the present study. Yet, the wording of the items should also ideally represent the appropriate level of analysis to which individual perception data will be aggregated [93], which is not the case in the present study, as items in the scale were formulated: “How often have you been confronted with the following… during the last six months?”. Hence, the findings should be replicated with appropriate level items.

There was a high correlation between the outcome and the moderator at the department level. However, our main research question was not whether the bullying rates of departments are related to a hostile climate in those departments. Rather, we focused on whether a contextual group-level factor (a hostile climate) moderates the relationship between work demands and exposure to bullying at work at the individual level (see also Figure 1 and Figure 2). Furthermore, the present study looks at a hostile climate only, and not a broad concept and measure of organisational climate. Hence, we look only at one characteristic of the prevailing organisational climate, which of course also may have other and even much more positive characteristics. Although our measure looks at the extent that the employees in the department are involved in either escalated interpersonal conflicts or being subjected to aggressive outlets from others, we lack detailed information on who or how many in the environment are behaving in an aggressive manner and who the opponents are in the perceived conflicts. There may be departments where there is mainly one aggressor, e.g., a manager who is misbehaving towards a range of subordinates, or one main escalated conflict involving many employees.

Further, scales on quantitative demands may be sensitive to the choice of specific items [94]. Kristensen and colleagues [94] argue that if items regarding fast work pace and tempo are included in a scale, several blue-collar jobs will be identified as high-demand jobs. While, on the other hand, items regarding long working hours and overtime will be more relevant for white-collar workers. As our sample consists of academics, hence mainly white-collar workers, it would have been interesting to include questions regarding working hours or whether they think they have time to finish their work tasks, to see whether this would affect the results. However, we did get significant results as hypothesized by using the workload scale in the present study, indicating that the items are not as irrelevant for our sample as argued by Kristensen and colleagues [94].

The present study findings also need further validation in other work contexts, as our sample only consists of employees in one Belgian university. Thus, the findings are not necessarily generalizable to all other occupational groups. Future studies should also include information on the number of perpetrators and the amount of social support received, as these variables were not included in the present dataset.

## 5. Conclusions

Given the scarcity of studies investigating the interaction of risk factors for bullying at different organizational levels, we believe the present study is important from the perspective of bullying prevention. Findings from the present study shed light on how work-related stressors interact with a hostile work climate in predicting exposure to bullying behaviours. Yet, our results also pinpoint that the role played by a hostile climate may vary between stressors, as a hostile climate played a more important role in relation to perceived role conflicts as compared to perceived high levels of workload.

Based on these results, we encourage both researchers and practitioners to continue to broaden their understanding of the antecedents of workplace bullying by considering the different organizational levels. We believe that this more complex and integrated approach to exploring workplace bullying sets a strong foundation for future research and encourages researchers to further investigate the critical role that the organizational climate can play in accelerating or preventing workplace bullying.

## Figures and Tables

**Figure 1 ijerph-20-04464-f001:**
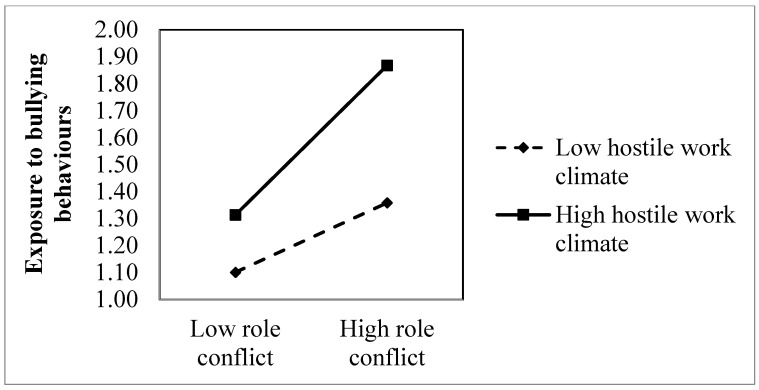
Plot of the interactive relationship of role conflict and bullying behaviours in departments with weak vs. a strong hostile work climate.

**Figure 2 ijerph-20-04464-f002:**
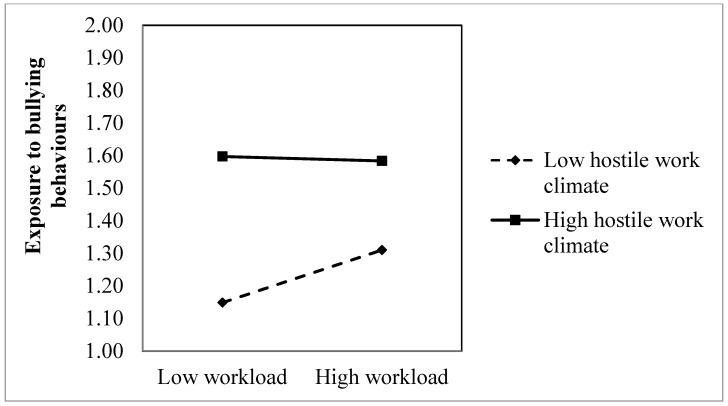
Plot of the interactive relationship of workload and bullying behaviours in departments with weak vs. a strong hostile work climate.

**Table 1 ijerph-20-04464-t001:** Mean, standard deviation, ICC, and within- and between-level correlations for all study variables (N = 1290 participants, N = 108 departments).

	$\bar{X}$	SD	ICC1/ ICC2	S^2^ between	S^2^ within	1.	2.	3.
*Within-level*								
1. Bullying behaviours	1.41	0.47	0.21 ^a^	0.05	0.19	-	0.27 ***	0.54 ***
2. Role conflict	1.62	0.49	0.13 ^a^	0.03	0.22	0.89 ***	-	0.55 *
3. Workload	1.56	0.66	0.04 ^a^	0.02	0.42	0.53 *	0.32 ***	-
*Between-level*								
4. Hostile work climate	0.24	0.17	0.13 ^a^ 0.85 ^b^	0.05	-	0.99 ***	0.53 *	0.86 ***

Note. Hostile work climate; ^a^ ICC1, within-level correlations; ^b^ ICC2, between-level correlations; Correlations above the diagonal are the correlations on the within-level. Correlations below the diagonal are correlations on the between-level. * *p* < 0.05, *** *p* < 0.001.

**Table 2 ijerph-20-04464-t002:** Multilevel estimates for the prediction of bullying behaviours.

Level and Variable	Null Model (Step 1)	Main Effect of L1 Predictors (Step 2)	Random Slope (Step 3)	Main Effect of L2 Predictor (Step 4)	Cross-Level Interactions (Step 5)
*Level 1*					
Intercept ($\gamma_{00})$	1.425 (0.024) ***	1.429 (0.025) ***	1.431 (0.024) ***	1.414 (0.013) ***	1.410 (0.012) ***
Role conflict ($\gamma_{10})$		0.474 (0.024) ***	0.466 (0.030) ***	0.469 (0.031) ***	0.454 (0.027) ***
Workload ($\gamma_{20}$)		0.071 (0.017) ***	0.065 (0.017) ***	0.068 (0.017) ***	0.060 (0.017) ***
*Level 2*					
Hostile work climate ($\gamma_{01})$				0.955 (0.065) ***	1.068 (0.068) ***
*Cross-level interaction*					
Role conflict ∗ Hostile work climate ($\gamma_{11})$					0.981 (0.153) ***
Workload ∗ hostile work climate ($\gamma_{12})$					−0.42 (0.115) ***
*Variance components*					
Within-unit (L1) variance ($\varepsilon_{0ij})$	0.193 (0.008) ***	0.135 (0.005) ***	0.128 (0.005) ***	0.127 (0.005) ***	0.126 (0.005) ***
Intercept (L2) variance ($\mu_{0j})$	0.040 (0.008) ***	0.051 (0.009) ***	0.051 (0.009) ***	0.006 (0.002) **	0.005 (0.002) *
Slope (L2) variance role conflict ($\mu_{1j})$			0.035 (0.0011) **	0.035 (0.012) **	0.014 (0.008)
Intercept-slope (L2) covariance role conflict			0.044 (0.008) ***	0.015 (0.004) ***	0.011 (0.003) ***
Slope (L2) variance workload ($\mu_{2j})$			0.000 (0.000)	0.000 (0.000)	0.000 (0.000)
Intercept-slope (L2) covariance workload			0.000 (0.000)	0.000 (0.000)	0.000 (0.000)
Loglikelihood	1666.63	1247.58	1194.92	1078.87	1041.57

Note. L1 = level 1; L2 = level 2; Robust standard errors of estimates are in parentheses. * *p* < 0.05, ** *p* < 0.01, *** *p* < 0.001.

## Data Availability

Data may be made available by contacting Guy Notelaers at the University of Bergen, Norway.

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
