# Peer review of "When the Going Gets Tough and the Environment Is Rough: The Role of Departmental Level Hostile Work Climate in the Relationships between Job Stressors and Workplace Bullying"

_ijerph, 2023, doi:10.3390/ijerph20054464_

Round 1

Reviewer 1 Report

Thank you for the opportunity to review this paper. The manuscript is well written, a strong sample was obtained, and the results are clearly presented. I also appreciated the use of a multi-level design, which are lacking within the bullying literature. However, I have some concerns about aspects of the manuscript which I set out below. Although some of these concerns are minor, I am quite concerned about the validity of the climate scale, and whether this actually assesses work climate, or something else.

Introduction

The work-environment hypothesis and the social interactionist approach to aggression are used to frame the study hypotheses. However, throughout the introduction, many of the arguments made are quite long, and convoluted, making the central thread quite difficult to grasp for the reader. Moreover, concepts from several other theoretical frameworks are used throughout the introduction, often without full explanation of what they entail, which further complicates the narrative e.g. the challenge-hinderance model, the JDR model, social information processing theory. The arguments made in this section could be made much more concisely by adopting a single overarching model, for example Ng et al., 2022, recently used the JD-R model to explain how the experience of bullying alongside passive / active-constructive bystanders results in worse outcomes for bullying targets. Could it also be the case that experiencing job demands within a hostile climate results in greater stress / frustration, and therefore greater likelihood of bullying perception? This suggestion may not prove workable, but I do think that some attention should be given towards clarifying and shortening this opening section, as currently it is not clear enough how workload and role conflict act as ambient stressors.

As a more minor comment, the paragraph starting on line 219 which refers to contributions would be better placed in the opening of the introduction section.

Method

Much more detail could be provided on the units and the sample in general. Although you note that the study takes place in a university, it is not clear what type of work occurs in these ‘units’ – are they work teams, or departments, people in the same building, or a mix of all these things?

I would also appreciate further information such as the average size and sd of members within units. Moreover, what was the rationale for removing units with less than 3 members?

My major concern in relation to this manuscript is the use of the hostile climate scale, which doesn’t appear to be a valid measure of climate. In the study, you state “A conflict management climate is defined as employees having confidence in that conflicts will be properly managed and resolved as in the organisation and its managers have proper procedures and routines for constructive conflict management (Einarsen et al., 2018; Rivlin, 2001).” And then later on “The organizational climate is based on employees perceptions of the policies, practices, and procedures as in climate of conflict management, and the behaviours they observe being accepted, rewarded and encouraged (Gruenert & Whitaker, 2015; Schneider et al., 2013), as in the case of the hostile working climate in the present study.” However, when examining the climate items, they bear no relation to the construct described above, indeed they don’t seem to measure work climate at all. The fact that these items come from a hazard scale and correlate so highly with bullying (.99) within this study is a matter of concern. Can you offer any statistical evidence of discriminant validity between these constructs e.g. Average variance extracted, CFA. If not, it seems very likely that the two measures are capturing the same construct, rather than different ones.

An additional point in relation to the above is that throughout the manuscript you use the terms ‘hostile climate’ and ‘climate for conflict management’ seemingly to refer to the same thing. It would be much clearer if you adopted a single term for the climate construct.

A final point on the climate measure is that it seems to assess aggression from other employees, but it makes no reference to whether these employees are from their work unit or not. It could be the case that the respondents are referring to aggressiveness from outside their unit. This point, alongside the one noted in the limitations section about capturing perceptions of the unit in general (rather than individual experiences) mean that the results are open to question.

Results

What was the ICC 1 of the hostile climate measure? It doesn’t seem to be included in the table.

Discussion

Although the discussion is generally well written, great emphasis is placed on the additive harmful effect of hostile climates. However, hostile climates only resulted in greater perceptions of bullying when considered in relation to role conflict, and it actually seemed to weaken the relationship between workload and bullying. With this in mind, the theoretical and practical implications of the discussion section should be tempered, as much more evidence is required before it can be stated that hostile climates enhance the likelihood of bullying perceptions.

Author Response

The work-environment hypothesis and the social interactionist approach to aggression are used to frame the study hypotheses. However, throughout the introduction, many of the arguments made are quite long, and convoluted, making the central thread quite difficult to grasp for the reader

Reply: We have now revised, partly rewritten and shortened the introduction in line with this comment hopefully making it a easier to follow. The last part of it has been moved to the theoretical part where we believe it will make more sense. We thank the reviewer for making us aware of the complexity of the introduction which have prompted this major revision. We are convinced that the introduction now reads much better. 

Based on this comment we have also revised the title of the manuscript to better capture the main aim and the problem addressed in this study.   

Moreover, concepts from several other theoretical frameworks are used throughout the introduction, often without full explanation of what they entail, which further complicates the narrative, e.g. the challenge-hinderance model, the JDR model, social information processing theory. The arguments made in this section could be made much more concisely by adopting a single overarching model, for example Ng et al., 2022, recently used the JD-R model to explain how the experience of bullying alongside passive / active-constructive bystanders results in worse outcomes for bullying targets. Could it also be the case that experiencing job demands within a hostile climate results in greater stress / frustration, and therefore greater likelihood of bullying perception? 

Reply: Again, we have revised the introduction in order to simplify the argument somewhat and while also relying more explicitly on the work environment hypothesis and the social interactionist perspective for the overarching theoretical framework, and in addition strengthen the arguments in the direction proposed by the reviewer.  We have also removed what we now realize where somewhat confusing references to other theories and models. 

We tried to rely more on the job demands resource model as proposed by the reviewer, without finding that it actually contributed to the paper. The problem is that this study has a "demand-demand perspective" with no explicit resource. The absence of a contextual resource, however, taps into the notion of a resource passageway in COR theory. And a hostile climate resonates well with that notion. Hobfoll defined a resources passageway not only as something positive : …organizational ‘environmental conditions that support, foster, enrich, and protect the resources of individuals, sections or segments of workers, and organizations in total, or that detract, undermine, obstruct, or impoverish people’s or group’s resource reservoirs. Hence, we have argued that a hostile climate indicates less available social resources and therefore not only being something which indicates a loss of resources but something which acts as a demand (Schaufeli & Taris. 2013).. We believe that by focusing this notion of resource passageway and on COR theory we have hopefully addressed this concern by the reviewer.  

My major concern in relation to this manuscript is the use of the hostile climate scale, which doesn’t appear to be a valid measure of climate. In the study, you state “A conflict management climate is defined as employees having confidence in that conflicts will be properly managed and resolved and that the organization and its managers have proper procedures and routines for constructive conflict management (Einarsen et al., 2018; Rivlin, 2001).” And then later on “The organizational climate is based on employees perceptions of the policies, practices, and procedures as in climate of conflict management, and the behaviours they observe being accepted, rewarded and encouraged (Gruenert & Whitaker, 2015; Schneider et al., 2013), as in the case of the hostile working climate in the present study.” However, when examining the climate items, they bear no relation to the construct described above, indeed they don’t seem to measure work climate at all.

An additional point in relation to the above is that throughout the manuscript you use the terms ‘hostile climate’ and ‘climate for conflict management’ seemingly to refer to the same thing. It would be much clearer if you adopted a single term for the climate construct.”

Reply: We thank the reviewer for making us aware of the seemingly confusion created in the introduction on these matters. We have therefore rewritten and revised the introduction to make it clear that what constitutes a hostile work climate which we are looking at and that this is not at all a measure of Climate for conflict management, quite contrary. In addition we have added a parallel study on the moderating effect of high-performance HRM systems recently publiseed to be more fair to the litterature and to make it even more clear that these are other but relevant concepts.  These concept are only included to show earlier research on climate and bullying with some empirical evidence relevant for our study. We have defined our climate concept better and more explicit, and also explained better our purpose and what we are looking at. We do think that our measure and the items involved are highly relevant in relation to this climate concept as we aggregate the perception made by all in the said department of the level of  interpersonal conflicts and aggressiveness. Hence, we create an variable that describes the general climate in the department and how this is permeated by escalated interpersonal conflicts and aggressiveness, what we coin a hostile climate.  

The fact that these items come from a hazard scale and correlate so highly with bullying (.99) within this study is a matter of concern. Can you offer any statistical evidence of discriminant validity between these constructs e.g. Average variance extracted, CFA. If not, it seems very likely that the two measures are capturing the same construct, rather than different ones.

REPLY: We understand your concern regarding any overlap between hostile climate and the level of bullying on a department level,  but our main research question is not whether bullying rates of departments are related with a hostile climate in these departments.  Rather, we focus on whether a contextual group level factor, in our case a hostile climate, can moderate the relationship between predictors (in our case two work demands) and outcome (exposure to bullying at work) at the individual level.  We therefore do not see that there is a need to perform a multilevel CFA to illustrate that both aggressive climate (AC) and workplace bullying climate (WBC) are distinct concepts because we do nowhere aim to explain whether AC is predicting WBC in our study.  With our focus on the individual level relationships, we try to find out if contextual factors can influence these relationships. In order to do so, we modelled a random slopes model (building upon a random intercepts model) with a cross level interaction that depicts the influence of a hostile climate on an aggregated level on the relationship between predictors and outcome on and individual level. Hence we look at whether individuals who experience job stressors will reported more exposure to bullying when being in a department where many other employees face interpersonal conflicts and aggression. Furthermore, Notelaers and colleagues (2018) using these items on an individual level, have shown that the exposure to bullying and involvement in escalated interpersonal conflicts with aggressive outlets are different factors yet with overlap when there is little exposure to both. 

I would also appreciate further information such as the average size and sd of members within units. Moreover, what was the rationale for removing units with less than 3 members?

Reply: We have now addressed these issue better in the method section and provided an argument for our procedure here, reading: “We only retained departments consisting over 3 respondents to secure a reasonable measure of a departments level hostile climate and reduce the risk of having only targets or perpetrators rating the climate.”

Much more detail could be provided on the units and the sample in general. Although you note that the study takes place in a university, it is not clear what type of work occurs in these ‘units’ – are they work teams, or departments, people in the same building, or a mix of all these things?

Reply: the units in this study are university departments and their formal  counterparts within  the administration and technical area, which is now more clearly stated throughout the article. The problem is that these formal units in the administrative and technical area are not necessarily denoted as a department, yet still constitute the formal organization of these parts of the University.  Hence, this is a heterogenous sample when it comes to work task and roles and there may of course be more informal work teams again within departments. However, departments are the most basic and the most formal organizational entities of work and employment in Universities as in most organizations. We thank the reviewers for this comment and believe that this is now more clearly described in the manuscript. 

Results

What was the ICC 1 of the hostile climate measure? It doesn’t seem to be included in the table.

Reply: The ICC1 for hostile climate was .125 and is now written as .13 in the table. Thank you for making us aware of this missing value.

Although the discussion is generally well written, great emphasis is placed on the additive harmful effect of hostile climates. However, hostile climates only resulted in greater perceptions of bullying when considered in relation to role conflict, and it actually seemed to weaken the relationship between workload and bullying. With this in mind, the theoretical and practical implications of the discussion section should be tempered, as much more evidence is required before it can be stated that hostile climates enhance the likelihood of bullying perceptions.

Reply: We appreciate this comment and do see your point. We have now revised the discussion part somewhat to accommodate this feedback.

Reviewer 2 Report

In general the document has been well written. In the same way, the sections shown as well as the information in each of these and the references used are appropriate. In addition, the proposed hypotheses are well supported in the literature and the methodology used for their verification has also been well described. However, here are some minor comments that should be addressed:

The research gap that is intended to be addressed should be better stated in the introduction.

In lines 355 and 387 the abbreviation n.s. appears, however, what it means in the text (not significant) was not directly declared.

In their document they declare 2 figures and refer to them, however, only one of them appears. Add the missing figure.

I believe that once these points are addressed, the work will be ready for publication.

Author Response

The research gap that is intended to be addressed should be better stated in the introduction.

REPLY: We have tried to make this more clear in the revised version.

In lines 355 and 387 the abbreviation n.s. appears, however, what it means in the text (not significant) was not directly declared.

REPLY: We have now made this clear in the sentence (originally in line 355) and thank the reviewer for this comment. However in line 387 this is is already included in the sentence as it states clearly that the simple slope test was not significant.

In their document they declare 2 figures and refer to them, however, only one of them appears. Add the missing figure.

REPLY: Thank you so much for pointing this out. Do not quite understand what happened here. However, we have sent that figure to the journal. It was not possible for us in the review-process to include it. Hence, this must be done by the journal. 

I believe that once these points are addressed, the work will be ready for publication.

Reply: We thank you for your positive evaluation of our paper. Yet, we hope that the paper is now even better argued and even better written after following the comments of the other reviewer.

Round 2

Reviewer 1 Report

Thank you for the effort that you put into the revision. I believe this manuscript is much better as a result. However, I still have some concerns about the climate measure and I am not of the view that you are measuring the organisational climate, but rather aggregated perceptions regarding whether people have been subjected to mistreatment. This may capture some aspects of the organisational climate as you argue, but given your focus on units it could also be simply assessing whether there is an individual within these units who engages in regular negative interpersonal behaviour towards others. In which case, the actual climate may be relatively positive, but the presence of a malevolent individual may be driving the responses to your scale. I think that this should be acknowledged in the manuscript and I also think that much greater effort needs to be invested in toning down some of the claims made in the discussion section, as there are alternative explanations for the findings presented. 

Author Response

Based on your valid comment on what may constitutes and cause the existing hostile climate and the need to tone down the claims made in the discussion, we have now included the following in the imitations part of the article to accommodate this concern:

"Also, the present study looks at hostile climate only, and not a broad concept and measure of organisational climate. Hence, we look only at one characteristic of the prevailing organisational climate, which of course also may have other and even much more positive characteristics. Although our measure looks at the extent that the employees in the department are involved in either escalated interpersonal conflicts or being subjected to aggressive outlets from others, we lack detailed information on who or how many in the environment that are behaving in an aggressive manner and who the opponents are in the perceived conflicts. There may be departments where there are mainly one aggressor, e.g. a manager who are misbehaving towards a range of subordinates, or one main escalated conflict involving many employees."

    Furthermore, to clarify better our findings and to tone down and make the needed nuances in our conclusions we have included the following sentence in the conclusion part:

 "Yet, our results also pinpoints that the role played by a hostile climate may vary between stressors as a hostile climate played a more important role in relation to perceived role conflicts as compared to perceived high levels of work load" 

We again thank the reviewer for this useful comments and for helping us in improving our article further.